# Sex Differences in the Impact of Dynapenic Abdominal Obesity on Mild Cognitive Impairment among Korean Elderly in the Community

**DOI:** 10.3390/healthcare12060662

**Published:** 2024-03-15

**Authors:** Jiyun Kim, Yuna Paik

**Affiliations:** 1School of Nursing, Gachon University, Incheon 21936, Republic of Korea; jkim@gachon.ac.kr; 2Department of Nursing, Seoil University, Seoul 02192, Republic of Korea

**Keywords:** dynapenic abdominal obesity, mild cognitive impairment, handgrip strength, waist circumference, dynapenia, sex differences

## Abstract

Abdominal obesity (AO) and dynapenia (DP) are associated with cognitive decline, and the relationship between dynapenic abdominal obesity (DAO), a combination of DP and AO, and mild cognitive impairment (MCI) has been confirmed. This study aims to determine whether this relationship exhibits potential sex differences. The relationship between MCI and DAO was confirmed in 1309 community elderly individuals aged 65 years or older who were not diagnosed with dementia. The MCI was defined as a Korean mini-mental state examination (K-MMSE) score of 18–23 points. Multiple logistic regression analyses were conducted, categorizing participants into groups: a control group without AO or DP, an AO group, a DP group, and a DAO group. The study results showed that in women, both DP and DAO were significantly associated with MCI not only in the unadjusted Model 1 but also in Model 2, which adjusted for general characteristics and health behaviors, and Model 3, which additionally adjusted for chronic diseases and disease-related characteristics. In men, DP was associated with MCI in the unadjusted Model 1. The findings highlight sex differences in the impact of the DAO on MCI. These differences should be considered when studying the factors related to MCI in old age.

## 1. Introduction

Cognitive function is an important indicator of mental health and successful aging in later years [1]. Aging is characterized by a decline in muscle mass and an increase in fat mass—essentially, the muscle becomes converted to fat [2]. Alterations in body composition are inherent to the aging process, with shifts in fat and muscle mass holding paramount significance [3]. Recent studies investigating factors associated with cognitive decline in the elderly suggest that obesity increases the risk of cognitive decline (MCI) [4,5], while regular physical activity reduces the risk of cognitive decline and dementia [6,7]. Longitudinal studies on aging suggest that maintaining or increasing muscle mass does not prevent age-related decline in strength [8,9]. Furthermore, dynapenia (DP) is defined as the age-related decline in muscle strength [10]. In previous studies, research has linked DP to a decline in cognitive function [11], and abdominal obesity (AO) in the elderly has been associated with an elevated risk of cognitive impairment and dementia, irrespective of body mass index (BMI) [12,13]. Dynapenic AO (DAO), a combination of DP and AO, has been shown to exert an additional deteriorating impact on cognitive function compared to either sarcopenia or obesity alone [14].

Meanwhile, older women engage in far less physical activity than men in the same age group, and older women have higher body fat and relatively lower muscle strength than men [15]. When elderly women become obese, even a small decrease in muscle strength can lead to difficulties with mobility or weight-bearing, and the resulting consequences become more serious [16]. In a previous study, it was demonstrated that DAO was related to MCI. However, sex differences were not confirmed because the analysis was not categorized by sex in that study [14]. Considering these characteristics in women, this study aims to determine the relationship between AO, DP, DAO, and MCI by considering differences in sex and analyzing how these relationships differ.

## 2. Materials and Methods

### 2.1. Subjects

This study analyzed the epidemiological survey data from the 6th follow-up (7th period) of a community-based cohort. These data are part of the Korean Genome and Epidemiology Study (KoGES), a large prospective cohort study conducted by the National Institute of Health (NIH), the Korea Disease Control and Prevention Agency, and the Republic of Korea [17]. These cohort data were established in 2001 and are being tracked until 2020. The data utilized in this analysis were the 6th follow-up data, encompassing 5906 individuals who were surveyed between 2013 and 2014. To access data for analysis, the researcher applied for data use at NIH after receiving a research ethics review of the research plan at the affiliated institution, Gachon University (IRB number: 1044396-202203-HR-066-02). After passing NIH’s review, the researcher went to the data analysis room of the National Biobank of Korea and conducted data analysis. Among these subjects, those who were aged 65 years, had not been diagnosed with dementia or received treatment for depression, and had handgrip strength and BMI data were included. The analysis excluded patients with dementia with a Korean mini-mental state examination (K-MMSE) score of 17 or less. The final analysis utilized data from 1309 participants (597 men and 801 women) (Figure 1).

### 2.2. Definition of AO, DP, and DAO

Abdominal obesity was categorized based on waist circumference (WC). According to the results of previous studies on the WC of Korean adults, AO was defined as 90 cm or more for men, and AO was defined as 85 cm or more for women [18]. The DP is defined as low muscle strength measured by handgrip strength (HGS). The HGS was measured utilizing a digital hand dynamometer (digital grip strength dynamometer; T. K. K. 5401, Takei Scientific Instruments Co., Ltd., Nigata, Japan). If the subject underwent surgery or was injured in the hand or wrist within three months, only the uninjured hand was measured [19]. The HGS was measured with the subject sitting on a chair, placing his or her arm on a desk, and maintaining the arm angle at 90° [19]. The test was repeated for approximately 15 s. When the measurements of one hand are complete, the other hand is also measured using the same procedure [19]. As in previous studies, the average HGS of the dominant arm was utilized as the HGS value [20]. Low muscle strength was defined as HGS < 28 kg in men and <18 kg in women [21]. Participants were divided into four groups, namely: AO only, DP only, DAO with both, and Control without both [22].

### 2.3. Definition of Cognitive Impairment

The K-MMSE utilizes the cumulative score of all items, with a total score of 23 generally employed as the evaluation standard (cutoff point) for cognitive dysfunction. Epidemiological research findings suggest that, depending on the total K-MMSE score, scores of 24–30 indicate no cognitive impairment, scores of 18–23 indicate MCI, and scores of 0–17 indicate definite cognitive impairment [23]. Consequently, participants with a score of 17 or less were excluded from the study, and subjects with scores of 18–23 and 24–30 were divided into MCI and normal groups.

### 2.4. General Characteristics, Health Behavior, and Disease-Related Characteristics

General characteristics, such as age, sex, education level, and marital status, were measured using questionnaires. The BMI is calculated as weight divided by the square of the height. Among the health behaviors, smoking, drinking, and physical activity were also measured. Physical activity was classified into inactive, moderate, and active exercise groups by calculating the total amount of exercise per week as metabolic equivalent (MET) based on the International Physical Activity Questionnaire [24]. The inactive exercise group is less than 600 MET per week, the moderate exercise group is 600 to 3000 MET, and the active group is more than 3000 MET [24]. Weekly exercise is classified into 3.3 MET for walking, 4.0 MET for moderate-intensity exercise, and 8.0 MET for vigorous exercise, depending on the type of daily exercise [24]. The activity intensity was converted to MET by multiplying the number of participation days per week by the average activity time.

Disease-related characteristics included whether patients were diagnosed with hypertension, hyperlipidemia, ischemic heart diseases, or diabetes. Systolic blood pressure was measured, and blood examinations included total cholesterol, high-density lipoprotein (HDL) cholesterol, triglyceride (TG), serum high-sensitivity C-reactive protein (hsCRP), hemoglobin, and plasma glycohemoglobin (HbA1c).

### 2.5. Statistical Analysis

The characteristics of all participants are described, along with the mean and standard deviation (SD) and frequency percentages by sex. The sex differences in characteristics were analyzed with a *t*-test or chi-square test.

DAO groups such as control, AO, DP, and DAO were compared with the normal group and MCI group by sex with a chi-square test. HGS and WC were compared with the normal group and MCI group by sex with a *t*-test.

Odds ratios for MCI in the group of DAO categories (control, AO, DP, and DAO) in multiple regression analyses according to sex were calculated. Multiple logistic regression analyses were performed with Models 1, 2, and 3. Model 1 was the unadjusted model, while Model 2 was adjusted to the characteristics of the study subjects, including health behavior. Additionally, Model 3 was adjusted by including variables reflecting diseases that are frequent in the elderly. Furthermore, Model 2 was adjusted with age, education, marital status, smoking status, drinking status, BMI, and physical activity. Model 3 was adjusted with the variables in Model 2 and further adjusted in Model 2 for hypertension, dyslipidemia, ischemic heart disease, diabetes mellitus, SBP, total cholesterol, HDL cholesterol, TG, hsCRP, hemoglobin, and HbA1c. Multiple logistic regressions with the dependent variable HGS (low handgrip as HGS < 28 kg in men and <18 kg in women) and WC (high WC as 90 cm or more for men and as 85 cm or more for women) were carried out for Models 1, 2, and 3. All analyses were conducted using SAS version 9.4 (SAS Institute Inc., Cary, NC, USA). The statistical significance of all comparison values was based on *p* < 0.05.

## 3. Results

### 3.1. Characteristics of All Participants According to Sex

Participants characteristics by sex are presented in Table 1. Men were 72.3 years old, and women were 72.5 years old. Regarding the level of education, the proportion of women with no education was higher than that of men, and the proportion of middle school graduates or higher was higher among men than women. Marital status, smoking status, drinking status, BMI, and physical activity all differed between men and women. Specifically, the proportion of married people was higher in men, and the BMI was higher in women. There were more men than women who smoked and drank, and the rate of inactivity was higher among women than men. In terms of health-related characteristics, there were differences in SBP, total cholesterol, hsCRP, and hemoglobin between men and women. The average SBP and total cholesterol values were higher in women than in men, and the hsCRP and hemoglobin values were lower in women than in men.

### 3.2. DAO Parameters of the Normal Group and MCI Group According to Sex

The results of analyzing whether there were differences in the DAO category, HGS, and WC between the normal cognitive group and MCI according to sex are presented in Table 2.

There was no difference in frequency between the normal cognitive group and MCI according to the DAO category (control, AO, DP, and DAO) in men, but there was a difference in women. There was a difference in handgrip strength values between the normal cognitive group and MCI in men and women. There was no difference in WC values between the normal cognitive group and MCI in men and women.

### 3.3. Odds Ratios for MCI in the Group of DAO Category in Multiple Logistic Regression Analysis According to Sex

Multiple logistic regression results obtained by calculating the odds ratio according to sex, using control as a reference among DAO categories, are presented in Table 3. In Model 1, an unadjusted model, DP was significantly associated with MCI in men (OR = 1.79, 95% CI = 1.01–3.16). In women, the odds for DP compared to control were 3.00 (95% CI = 1.71–5.26), and for DAO, the odds were 2.21 (95% CI = 1.36–3.59). In Models 2 and 3, no significant odds values were calculated for each man in the DAO category. In Model 2, for women, both DP and DAO were significantly associated with MCI (OR = 2.29, 95% CI = 1.22–4.34, OR = 2.15, 95% CI = 1.18–3.96, respectively). Similarly, in Model 3, for women, both DP and DAO were significantly associated with MCI (OR = 2.42, 95% CI = 1.28–4.64, OR = 2.37, 95% CI = 1.28–4.45, respectively).

### 3.4. Association between Handgrip Strength and High Waist Circumference in the Logistic Analysis According to Sex

Table 4 shows the results of a multiple logistic regression analysis that categorized HGS and WC and calculated the odds ratio by sex. The odds ratio for the likelihood of developing MCI for low HGS was calculated to be significant only in Model 1 for men. (odds ratio = 1.70, 95% CI = 1.12–2.59). For low HGS, significant odds ratios were calculated in Models 1, 2, and 3 for women. No significant odds ratios were calculated for both men and women when WC was high compared to when WC was low.

## 4. Discussion

This study aimed to determine the relationship between the DAO category (control, AO, DP, and DAO) and MCI according to sex in community-dwelling elderly individuals. The results of the study exhibited that DP and DAO were related to MCI in women, not only in the unadjusted Model 1 but also in Model 2, which adjusted for general characteristics and health behaviors. Additionally, Model 3 adjusted for chronic disease and disease-related characteristics. In men, DP was associated with MCI in the unadjusted Model 1. Previous studies have exhibited that elderly women with low muscle strength (<20.4 kg) have a 3.04 times higher risk of being exposed to MCI than participants with high physical function [25]. It was confirmed that there is an independent relationship with the risk of sarcopenic MCI in elderly women [25]. In cross-sectional studies, MCI has been shown to be associated with lower HGS [26].

When DAO was divided into groups, the relationship with MCI was significantly greater when DP and DAO were utilized than when AO was utilized alone, suggesting that muscle strength or muscle strength and obesity at the same time are simultaneously more related to MCI than simple obesity. These results are consistent with those of previous research exhibiting that cognitive function and muscle strength are related [11,12,27].

As age increases, physical function and muscle function decrease [28], which is related to a decline in cognitive function, and strength as opposed to muscle mass is related to cognitive decline [11,25]. High skeletal muscle power reduces the risk of frailty, cardiovascular disease, cardiovascular disease-related factors, and mortality [29]. The relationship between cognitive function and low muscle strength can be inferred from changes in nervous system activity and white matter integrity, such as weakened muscle strength or decreased frontal lobe function, which requires neuromuscular coordination [30]. In a cohort study examining the relationship between HGS and neuroimaging, brain white matter hyperactivity was significantly greater when HGS was low [31]. These findings reflect the research results demonstrating that vascular endothelial dysfunction causes cognitive aging [32]. Additionally, this is related to muscle weakness [33].

Muscle strength was measured using the HGS, which can be easily measured in community and clinical settings [34]. Additionally, it is employed as a diagnostic criterion for sarcopenia and exhibits a high correlation with overall cognitive functions such as executive function, attention, and language [3]. The decrease in HGS exhibits a significant inverse correlation with an individual’s cognitive function; therefore, it is a functional marker that can infer the degree of cognitive decline and is a crucial indicator primarily utilized in muscle strength evaluation in sarcopenia [3].

In a previous study, a relationship between men’s handgrip power and MCI was confirmed [26]. However, in this study, a weak relationship was confirmed solely in the univariate analysis. In a cohort study that followed a large population over a relatively long period of time, the relationship between HGS and dementia risk, cognition, and neuroimaging was confirmed, and similar results were found in men and women [31]. Conversely, there is a previous study that found differences in the relationship between sarcopenia and MCI depending on sex [35]. In this study, the relationship between sarcopenia and MCI according to sex was due to sex hormones such as androgen and estrogen [35]. In other words, there is a difference in the distribution of sex hormone receptors that affect muscle mass in the hippocampus between men and women [36]. Since differences between men and women in the relationship between muscle propensity and MCI are not consistently shown, there is a need to conduct long-term studies targeting both male and female populations. In addition, it is necessary to assess other muscle strength parameters that can predict cognitive decline, such as gait speed, step length, and timed chair stand test results [37].

Considering the changes in body composition during old age, caution is required when interpreting the relationship between obesity in old age and cognitive function. BMI is one of the most commonly employed obesity indices; however, it has limitations in reflecting the amount of fat in the body and does not effectively reflect body types with several muscles or body types with little muscle or fat [13]. Even if body weight is maintained in old age, body composition changes. Age-related changes in body composition, particularly decreased muscle mass and increased body fat, strongly interact with each other from an etiological perspective [38]. Visceral fat is positively correlated with CRP; therefore, low-intensity inflammation is believed to be an important cause of sarcopenic obesity [38]. In addition to inflammatory factors, insulin resistance also plays a major role in the development of sarcopenic obesity. Insulin resistance is independently related to muscle weakness, whereas resistance exercise improves muscle insulin resistance [39], and low physical activity is an important risk factor considered for weight gain. Obese individuals have reduced physical activity, which increases the likelihood that it will lead to decreased muscle strength [40].

This process generates DAO in the elderly, which is an important health problem, acting as a mediating factor that changes brain structure and function and affects cognitive function [5,14]. In this study, no significant relationship was found between abdominal circumference alone and MCI. Adiposity is a well-known risk factor for dementia. A recent study exhibited that the normal-weight group with abdominal obesity had a significantly increased risk of dementia compared to the group without abdominal obesity [13]. However, in previous studies, general obesity was not associated with dementia, but central obesity was positively correlated [41]. In a previous study that confirmed the degree of prediction of late-onset Alzheimer’s disease (LOAD) by measuring WC and waist-to-hip ratio (WHR) as indicators of central obesity in the elderly, the highest WC quartile was compared between LOAD and LOAD in a model adjusted for age and sex [41]. There was a relationship; however, in the model adjusted for other covariates, this relationship was significantly reduced [41]. The highest WHR quartile was most strongly associated with LOAD, and this association was robust in all models, leading researchers to suggest the utilization of WHR as a measure of central obesity [41].

As sarcopenic obesity not only affects physical activity, metabolism, and cardiovascular function but also cognitive function, research on it is of significant interest not only in medicine, nutrition, and geriatrics but also in public health. The treatment of sarcopenic obesity should focus not only on reducing body fat but also on maintaining and increasing strength, muscle function, and muscle mass [38]. There are also research results exhibiting that combined exercise with a variety of aerobic and resistance training exercises can be effective in reducing visceral fat and increasing muscle mass and strength to prevent the progression to dementia [42]. It is essential to develop exercise programs for the elderly population in the future.

This study has some limitations. A cross-sectional analysis was performed to determine the relationship between the DAO and MCI. Therefore, care must be taken to check the relationship between the DAO and MCI. In the future, a cohort data analysis is required to confirm the relationship between the DAO status and MCI occurrence. As this study utilized Koreans as the standard for AO to define DAO, and in defining MCI, the K-MMSE score was classified, caution should be exercised when comparing populations in other countries. In addition, the subjects of this study were elderly individuals in certain regions of Korea; therefore, it is difficult to regard them as representative of the Korean elderly. Nevertheless, this study confirmed that when considering the characteristics of obesity related to MCI in elderly women, muscle strength or muscle strength and obesity should be considered simultaneously, as opposed to AO alone.

## 5. Conclusions

Sex differences in the influence of the DAO on MCI were observed, and these distinctions should be considered when formulating strategies for preventing cognitive decline. Future research is warranted to elucidate sex-specific differences in the incidence of MCI, encompassing forms of obesity, muscle strength, muscle function, and muscle mass.

## Figures and Tables

**Figure 1 healthcare-12-00662-f001:**
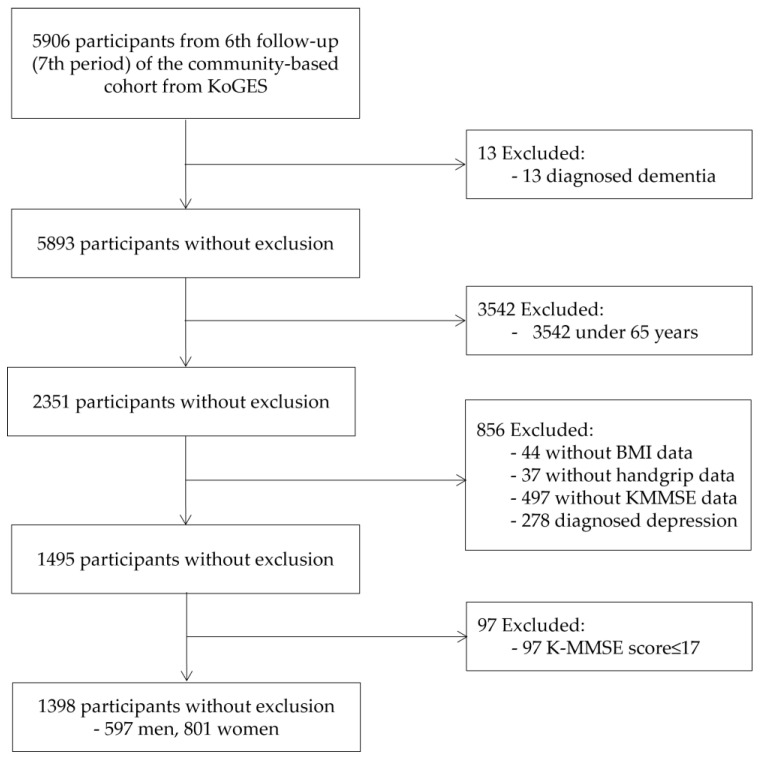
Flow chart illustrating the inclusion of participants. KoGES: Korean genome and epidemiology study; BMI: body mass index; K-MMSE: Korea-Mini Mental State Examination.

**Table 1 healthcare-12-00662-t001:** Characteristics of all participants according to sex (*N* = 1398).

Variable	Groups	Men	Women	*p*
Age (mean, SD, year)		72.3 (4.7)	72.5 (4.6)	0.493
Age (n, %)	65~69	204 (34.2)	238 (29.7)	0.207
	70~79	312 (52.3)	446 (55.7)	
	80 and over	81 (13.6)	117 (14.6)	
Education (n, %)	No school	31 (5.2)	158 (19.7)	<0.001
	Elementary school	43 (7.2)	182 (22.7)	
	Middle school and over	523(87.6)	461 (57.6)	
Marital status (n, %)	Divorced/bereaved, etc.	43 (7.2)	341 (42.6)	<0.001
	Married	554 (92.8)	460 (57.4)	
BMI (mean, SD, kg/m^2^)		23.5 (3.1)	24.6 (3.5)	<0.001
Smoking status (n, %)	Non-smoker/past smoker	461 (77.2)	793 (99.0)	<0.001
	Current smoker	136 (22.8)	8 (1.0)	
Drinking status (n, %)	Non-drinker/past drinker	258 (43.2)	688 (85.9)	<0.001
	Current drinker	339 (56.8)	113 (14.1)	
Physical activity (n, %)	Inactive	426 (71.4)	648 (80.9)	<0.001
	Moderate	140 (23.5)	146 (18.2)	
	Active	31 (5.2)	7 (0.9)	
Hypertension (n, %)	Yes	33 (5.5)	41 (5.1)	0.736
	No	564 (94.5)	760 (94.9)	
Dyslipidemia (n, %)	Yes	25 (4.2)	43 (5.4)	0.310
	No	572 (95.8)	758 (94.6)	
Ischemic heart diseases (n, %)	Yes	8 (1.3)	17 (2.1)	0.275
	No	589 (98.7)	784 (97.9)	
Diabetes mellitus (n, %)	Yes	17 (2.9)	17 (2.1)	0.384
	No	580 (97.1)	784 (97.9)	
SBP (mean, SD, mmHg)		127.4 (16.3)	130.5 (17.5)	<0.001
Total cholesterol (mean, SD, mg/dL)		169.1 (31.9)	178.0 (32.5)	<0.001
HDL cholesterol (mean, SD, mg/dL)		48.8 (13.9)	49.6 (11.8)	0.223
TG (mean, SD, mg/dL)		123.6 (80.8)	131.4 (66.6)	0.056
hsCRP (mean, SD, mg/L)		2.2 (5.2)	1.6 (3.2)	0.008
Hemoglobin (g/dL)		14.2 (1.3)	12.7 (1.0)	<0.001
HbA1c (%)		5.8 (0.8)	5.9 (0.7)	0.169
Handgrip strength (mean, SD, kg)		30.2 (7.5)	17.7 (4.2)	<0.001
Waist circumference (mean, SD, cm)		88.8 (9.0)	90.3 (9.4)	0.003
DAO category (n, %)	Control	188 (31.5)	100 (12.5)	<0.001
	AO	189 (31.7)	307 (38.3)	
	DP	130 (21.8)	115 (14.4)	
	DAO	90 (15.1)	279 (34.8)	

SD, standard deviation; BMI, body mass index; SBP, systolic blood pressure; HDL, high-density lipoprotein; TG, triglyceride; hsCRP, high-sensitivity C-reactive protein; HbA1c, glycated hemoglobin; DAO, dynapenic abdominal obesity; independent *t*-test or chi-square test.

**Table 2 healthcare-12-00662-t002:** DAO parameters of the normal group and the MCI group according to sex (*N* = 1398).

DAO Category (n, %)	Men			Women		
	Normal	MCI	*p*	Normal	MCI	*p*
Control	160 (32.9)	28 (25.5)	0.095	69 (14.9)	31 (9.2)	<0.001
AO	159 (32.7)	30 (27.3)		205 (44.3)	102 (30.2)	
DP	99 (20.3)	31 (28.2)		49 (10.6)	66 (19.5)	
DAO	69 (14.2)	21 (19.1)		140 (30.2)	139 (41.1)	
Handgrip strength (mean, SD, kg)	30.8 (7.4)	27.5 (7.06)	<0.001	18.3 (4.0)	16.7 (4.3)	<0.001
Waist circumference (mean, SD, cm)	88.8 (8.9)	88.9 (9.81)	0.902	90.6 (9.2)	89.8 (9.6)	0.236

MCI, mild cognitive impairment; AO, abdominal obesity; DP, dynapenia; DAO, dynapenic abdominal obesity; independent *t*-test or chi-square test.

**Table 3 healthcare-12-00662-t003:** Odds ratios for MCI in the group of DAO category in multiple logistic regression analysis according to sex.

		Men	Women
		OR (95% CI)	OR (95% CI)
Model 1	Control	1.00	1.00
	AO	1.08 (0.62–1.89)	1.11 (0.68–1.80)
	DP	1.79 (1.01–3.16)	3.00 (1.71–5.26)
	DAO	1.74 (0.92–3.27)	2.21 (1.36–3.59)
Model 2	Control	1.00	
	AO	0.92 (0.45–1.90)	1.30 (0.71–2.41)
	DP	1.15 (0.61–2.16)	2.29 (1.22–4.34)
	DAO	1.13 (0.51–2.47)	2.15 (1.18–3.96)
Model 3	Control	1.00	1.00
	AO	0.93 (0.44–1.94)	1.41 (0.76–2.67)
	DP	1.23 (0.64–2.34)	2.42 (1.28–4.64)
	DAO	1.23 (0.54–2.75)	2.37 (1.28–4.45)

Model 1: Unadjusted model. Model 2: Adjusted for age, education, marital status, smoking status, drinking status, BMI, and physical activity. Model 3: Further adjusted for hypertension, dyslipidemia, ischemic heart disease, diabetes mellitus, SBP, total cholesterol, HDL cholesterol, TG, hsCRP, hemoglobin, and HbA1c. MCI, mild cognitive impairment; DAO, dynapenic abdominal obesity; OR, odds ratio; CI, confidence interval.

**Table 4 healthcare-12-00662-t004:** Association between handgrip strength or high waist circumference in the logistic analysis according to sex.

		Men	Women
		OR (95% CI)	OR (95% CI)
Low handgrip	Model 1	1.70 (1.12–2.59)	2.23 (1.68–2.98)
	Model 2	1.14 (0.72–1.81)	1.83 (1.33–2.53)
	Model 3	1.20 (0.74–1.91)	1.87 (1.35–2.60)
High waist circumference	Model 1	0.98 (0.65–1.49)	0.85 (0.62–1.16)
	Model 2	1.17 (0.75–1.84)	0.84 (0.59–1.19)
	Model 3	1.26 (0.77–2.06)	0.88 (0.61–1.28)

Model 1: Unadjusted model. Model 2: Adjusted for age, education, marital status, smoking status, drinking status, BMI, and physical activity. Model 3: Further adjusted for hypertension, dyslipidemia, ischemic heart disease, diabetes mellitus, SBP, total cholesterol, HDL cholesterol, TG, hsCRP, hemoglobin, and HbA1c. OR, odds ratio; CI, confidence interval.

## Data Availability

The data analyzed in this study are de-identified data from the KoGES research and are publicly available upon request under the request policy available online.

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
