# Peer review of "Sex Differences in the Impact of Dynapenic Abdominal Obesity on Mild Cognitive Impairment among Korean Elderly in the Community"

_healthcare, 2024, doi:10.3390/healthcare12060662_

Round 1
Reviewer 1 Report
Comments and Suggestions for Authors
This manuscript examined potential sex differences within relationships found between obesity conditions (i.e., AO, DP, DAO) and cognitive decline and or impairment. Findings from this work make an important contribution in identifying these differences in a Korean population and demonstrating the relevancy of tailoring interventions for people living with cognitive impairment and obesity according to biological sex differences. That said, there are a few areas where this manuscript could be strengthened by providing additional information, clarification, or support. The following provides more information.
Abstract
1. It would be more precise to use the phrasing “sex differences”, rather than sexual.
2. The last sentence says these differences should be considered for developing strategies for preventing cognitive decline. I am wondering about the accuracy of that sentence, especially considering there is no cure for dementia and other cognitive impairment conditions. Can you clarify how this information can be beneficial for interventions targeting older adults with either MCI, obesity or both?
Intro
3. The following statement is a little confusing:
“Early intervention for cognitive impairment plays a crucial role in slowing the progression to dementia; therefore, it is important to identify and address risk factors that contribute to initial cognitive decline, such as mild cognitive impairment.”
Can you clarify how impairment (ie a potential end result) contributes to decline (i.e. what leads to an end result)? The paper you reference seems to focus more on body composition and impairment, so it’s not as clear how you are using it here. Do mean that it’s important to address risk factors of cognitive decline including obesity?
4. Also, the discussion of research finding associations between DP and cognitive functioning decline and AO with cognitive impairment is interesting and brings up a question about why these conditions may be related to either decline or impairment. It could add a lot to the background of your work to have a sentence or two further explaining or offering theories as to why these relationships may exist. Is there literature suggesting a mechanism that links these conditions?
5. The following statement makes a generalization that is problematic:
“Meanwhile, older women engage in far less physical activity than men in the same age group, and older women have higher body fat and relatively lower muscle strength than men.”
There’s no citation to support either of these claims. Also, qualifying both statements, especially the first one, is important so that you aren’t stereotyping and suggesting all women engage in less physical activity than men.
6. The next part of this discussion is also a bit confusing:
“Therefore, there is a high possibility that it can only get worse because of obesity or muscle loss.”
It’s not clear what’s meant by “it can only get worse.” Do you mean women’s health can get worse because of obesity or muscle loss? That seems to imply there’s a high risk for all women to become obese. (This also brings the question does all muscle loss lead to obesity? In a sense, aren’t these quite separate physical changes?) Or do you mean cognitive decline is likely worse for women due to higher risk for obesity? If that’s the case, this claim needs more support and explanation of the link between obesity and cognition.
Regarding the section: 2.3 Definition of Cognitive Impairment
7. Understanding how links between health and cognition for men and women may vary is quite important. That said, it’s also important to be careful of language that can come across as stigmatizing, which can carry long ranging impacts on how professionals and families view and react to dementia and MCI. To classify one group as having MCI and then another group as “normal” suggests a devaluation or othering of people who are living with cognitive impairment. Is there another way you can characterize people who do not show symptoms of MCI? Maybe the group without MCI?
Results
8. When describing general characteristics and differences between men and women, the description of education is a little confusing. It’s stated: “Regarding education level, the proportion of middle school and over was higher in women and men, but men had a higher rate of high education.” Do you mean that most subjects had completed middle school and over, regardless of biological sex? There’s a mention of higher education, though this is not included in table 1. I’d recommend adding that in the table for greater support.
9. This description is followed by statements that other characteristics (e.g., marital status, physical activities, hemoglobin, etc.) showed significant differences between men and women. It would add precision to this section to indicate what that difference was in general, at least in regard to the health measures. (i.e., that women in this sample in general showed poorer health outcomes.)
Regarding the section starting “3.2 DAO parameters…”
10. It’s stated: “There was no difference in frequency between the normal cognitive group and MCI according to the DAO category in men, but there was a difference in women.” Yet, the table shows that for the DAO row, the difference for women in the two groups was 1 (140 vs 139). There’s no indication of a significant p value there as well. I’d clarify this discrepancy.
11. Also for Table 2: It could help clarify by labeling the columns that show p values. So the 4th and 7th columns should have a label in the header of p-value.
For the section “3.3 Odds ratios for MCI in the group of DAO…”
12. There are a few places where it’s phrased: “the odds of DP becoming MCI”. That implies a direct line from the diagnosis of DP to causing MCI (or the obesity diagnosis developing into MCI.) Overall though it seems this is more about odds of a greater association between the two health conditions. I’d recommend clarifying that meaning a bit more in the language.
Discussion
13. Strong discussion. There was still a question that might be considered in this section or earlier, namely, theoretical or empirical reasoning for a relationship between obesity and mental capability. Also, is there reasoning why these relationship are stronger in women? In thinking that regardless of how active an older adult is, if there is the presence of DAO, is there an explanation for why it’s a greater MCI risk for women men? Wondering if that question can be addressed.
Overall, I really enjoyed reading about your study. Thank you for this contribution!
Comments on the Quality of English Language
Overall the writing was clear, though there are some places where the phrasing was misleading and/or potentially stigmatizing. These are likely easy edits to how the authors are applying phrasing. I have specific examples in the above comments.
Author Response
"Please see the attachment."

Reviewer 2 Report
Comments and Suggestions for Authors
1- Generalizability: Elaborate on the generalizability of your findings to populations outside the Korean elderly group, considering cultural and geographic differences.
2- Sex Differences Discussion: Provide a deeper analysis of why significant sex differences were observed in the impact of dynapenic abdominal obesity on cognitive impairment.
3- Methodological Details: Offer more detail on the selection of statistical models used in your analysis.
Author Response
"Please see the attachment."

Reviewer 3 Report
Comments and Suggestions for Authors
This paper seems to address a reasonable question in a reasonable way. I have no particular concerns about specifics of the analyses or interpretation.
A couple of minor comments:
Throughout, the authors refer to "crude" models; I've never heard that term. I think they may mean "unadjusted "models (i.e., without covariates).
"sexual dimorphisms"; maybe 'sex differences' would be a better term.
Comments on the Quality of English LanguageMany parts of the manuscript are very well written, although I have some difficulty understanding their points in the discussion/interpretation of analyses.
Author Response
"Please see the attachment."
